# Can Genetic Markers Predict the Sporadic Form of Alzheimer’s Disease? An Updated Review on Genetic Peripheral Markers

**DOI:** 10.3390/ijms241713480

**Published:** 2023-08-30

**Authors:** Danelda Theron, Lloyd N. Hopkins, Heidi G. Sutherland, Lyn R. Griffiths, Francesca Fernandez

**Affiliations:** 1School of Behavioural and Health Sciences, Faculty of Heath Sciences, Australian Catholic University, Banyo, QLD 4014, Australia; danelda.theron@acu.edu.au; 2Centre for Genomics and Personalised Health, School of Biomedical Sciences, Queensland University of Technology, 60 Musk Ave, Kelvin Grove, QLD 4059, Australia; lloyd.hopkins@hdr.qut.edu.au (L.N.H.); heidi.sutherland@qut.edu.au (H.G.S.); lyn.griffiths@qut.edu.au (L.R.G.)

**Keywords:** Alzheimer’s disease, genes, micro-RNA, circular-RNA, long-coding RNA, biomarkers, peripheral tissues

## Abstract

Alzheimer’s disease (AD) is the most common form of dementia that affects millions of individuals worldwide. Although the research over the last decades has provided new insight into AD pathophysiology, there is currently no cure for the disease. AD is often only diagnosed once the symptoms have become prominent, particularly in the late-onset (sporadic) form of AD. Consequently, it is essential to further new avenues for early diagnosis. With recent advances in genomic analysis and a lower cost of use, the exploration of genetic markers alongside RNA molecules can offer a key avenue for early diagnosis. We have here provided a brief overview of potential genetic markers differentially expressed in peripheral tissues in AD cases compared to controls, as well as considering the changes to the dynamics of RNA molecules. By integrating both genotype and RNA changes reported in AD, biomarker profiling can be key for developing reliable AD diagnostic tools.

## 1. Introduction

With a worldwide demographic shift toward an aging population, dementia is becoming a growing public health concern. Alzheimer’s Disease (AD) is the most common form of dementia, affecting over 55 million individuals worldwide, with an estimated 10 million new cases per year (WHO, 2022). Symptoms of AD vary throughout the pathophysiological progression, starting from a preclinical phase (amyloid plaques and neurofibrillary tangle (NFT) accumulation in the brain) to mild and moderate cognitive impairment (progressive neuroinflammation and hippocampal shrinking), and final severe symptoms, including significant dysfunction of memory and executive functions, as well as personality disorders resulting from frontal and temporal neuronal loss [1]. Current interventions are most effective during the preclinical phase, with novel treatments being able to remove or lower pathogenic protein aggregation but not reverse existing neurodegeneration [2,3]. Consequently, it is critical to characterise early biomarkers for AD. In the last decade, various biomarkers have been tested in both peripheral human tissues and biofluids, such as cerebrospinal fluid (CSF), blood, saliva, nasal secretions, and urine [4,5,6,7,8,9,10,11,12]. Advances in neuroimaging techniques and peripheral biomarker testing greatly inform on the level of progression of the pathology including amyloid plaques, NFTs, neuroinflammation, microglial activation [13], neuronal injury [14] and vascular dysregulation [14]; however, these techniques fail to predict the potential presence of AD at the earliest stage. In addition, the assay specificity and sensitivity, costs and in some cases invasive nature of these tools restrict their use in clinical diagnostic and/or screening of the disease.

With recent advances in technology leading to lower costs, genetic screening has become increasingly utilised in the diagnosis of a range of diseases, including for the diagnosis of progressive neurological disorders such as Spino–Cerebellar Ataxia (SCA) [15]. Although the familial form of AD has strong genetic origins with well-characterised pathogenic variants (Amyloid Protein Precursor (APP), Presenilin-1 (PSEN1) and Presenilin-2 (PSEN2) [16,17]), the genetic landscape of sporadic AD is more complex and remains less well understood. Taking into account the multifactorial aetiology of AD (including environmental factors) [1], it is essential to consider not only the multiple genes involved in AD but also the regulation of the transcriptome. Small non-coding molecules, such as microRNAs, circular-RNAs, and long non-coding RNAs, can regulate gene expression at the transcriptional and post-transcriptional level. Consequently, it is essential to further explore the interplay between the genome and its regulation via non-coding RNAs, which may be influenced by and respond to environmental factors.

Keeping in mind the need for early biomarker detection in human peripheral tissues, and a lack of validated genetic screening tests, AD genetic susceptibility and regulation of RNA dynamics will be reviewed and discussed in the context of AD pathophysiology.

## 2. AD Genetic Susceptibility

A wide variety of genes implicated in the development of AD have been identified as possible diagnostic and therapeutic targets. Altogether, approximately 70% of sporadic AD’s heritability may be explained by genetic factors, including genes associated with multiple neuropathological events, as summarised in Table 1 and illustrated in Figure 1 [18,19,20]. Polymorphisms in these genes impact neuroinflammation, formation and clearance of abnormal/pathogenic proteins (Aβ peptides and phosphorylated-tau/tau), neural repair, and synaptic signalling. While not all variants are specific to AD, rare AD-related variants have been found to be significantly associated with lipid metabolism and amyloid processing [20].

### 2.1. Genes Related to AD Hallmarks of Pathophysiology

The development of amyloid plaques involves the interplay of several regulatory genes and AD pathways (Figure 1). The Amyloid-Precursor Protein (APP) gene encodes for the APP protein, which is cleaved by β-secretase (BACE) and γ-secretase to produce Aβ peptides. These peptides aggregate in excess in AD, forming characteristic amyloid plaques. Duplications or mutations of the β-cleavage site result in varying levels of pathogenic Aβ fragmentation, while mutations at different sites can shift the cleavage site, cause a conformational change, or alter protofibril and fibril formation [21]. Mutation near the BACE cleavage site, A673T (Icelandic), leads to a decrease in the production of Aβ40 [22]. In contrast, the A673V (A2V) mutation leads to increased pathogenic Aβ production [23]. Beyond the Aβ cleavage region (exon 16, 17), a recent study found a mutation in exon 5 (S198P) that leads to increased pathogenicity through Aβ cleavage by BACE1 [24].

**Table 1 ijms-24-13480-t001:** Mechanisms of AD Pathology and Associated Genes Reported by GWAS.

Mechanism of AD Pathophysiology	Genes
Amyloid Pathway Amyloid angiopathy	APP, ABCA7, ADAM10, BIN1, CD2AP, PICALM, SORL1, CR1, CLU, CD33, FERMT2, CASS4, PTK2B, SLC10A2, UNC5C, PLD3, SLC24A4, RIN3, DSG2, CST3, BCHE, CTSD, MEF2C, AB13, PLCG2, SHARPIN, APH1B, ACE, IGF1, INSR, LKB1, PSEN1, PSEN2, APOE, HLA-DRB5, HLA-DRB1
Tau Pathology	BIN1, CD2AP, PICALM, MAPT, CASS4, PTK2B, UNC5C, SLC24A4, RIN3, BCHE, HS3ST1, ACE, IGFQ, INSR, LKBQ
Lipid transport Lipid Metabolism	APOE, ABCA7, PLCG2A, SORL1, CLU, PICALM, SLC10A2, SLC24A4, RIN3, ECHDC3
Dendrites	CD2AP, COBL
Neuroinflammation microglial activation	CR1, TREM2, CLU, MS4A4E, ABCA7, CD33, MS4A6A, INPP5D, PLCG2, PTK2K, FBXL7, HLA-DRB5, HLA-DRB1, BIN1, MTHFR, MEF2C, AB13, SHARPIN, MINK1, ACE, SCIMP, PILRA, SPI1
Protein aggregation	APOE, PFDN1, CLU
Mitochondrial Function	MTHFD1L, ECHDC3, TOMM40
Synaptic dysfunction	PICALM, PTK2B, SLC10A2, MEF2C, MINK1, APH1B
Blood Brain Barrier disruption, vascular damage	CD2AP, EPHA1, MTHFR
Apoptotic genes	FBXL7, CLU
Oxidative Stress	MEF2C, NME8, TOMM40, MEF2C, MINK1, ACE
Cell Cycle and transport	BIN1, SORL1, PILCAM, CD2AP, EPHA1, RANDBP2, UNC5C, NME8, EPHA1, RANBP2, UNC5C, NME8, COBL, USP6NL, CD2AP

Summary of key genes associated with AD reported by GWAS replicated genetic studies. Adapted from [18,25,26].

In addition to the BACE gene, the Presenilin genes (PSEN1 and PSEN2) encode for key subunits of γ-secretase responsible for final APP cleavage into Aβ. These genes have been highly investigated, particularly for their role as risk factors for the familial form of AD [27]. Abnormal PSEN1 and PSEN2 result in the premature release of APP fragments during γ-cleavage, resulting in longer, incompletely proteolyzed amyloid fragments [27]. Functional polymorphisms (M139V, M146I, R278I) of PSEN1 lead to variability in PSEN1 protein lengths, γ-secretase function, and ratio of Aβ cleavages [28]. Mutations of D9E in PSEN1 remove a hydrophilic domain modifying the γ-secretase active site, while mutations at M139T impact APP binding [29]. Other mutations, including E280G, M146I and R278I, result in decreased auto-endo-proteolysis and larger PSEN1 protein lengths, destabilising γ-secretase function and resulting in a greater production of pathogenic Aβ [28,30]. In decreasing γ-secretase activity and carboxypeptidase activity, PSEN1 mutations are a key area of interest in understanding Aβ fragmentation patterns in AD [28]. While PSEN2 mutations are less common, they are still suspected to play a significant role in APP trafficking [31].

One of the primary genes influencing the risk of AD development is Apolipoprotein E (APOE). Three common APOE alleles (E2, E3, E4) have been well-characterised in various populations and may act in conjunction with various other AD risk factor genes [32,33]. The APOE2 allele has been reported as protective in AD, improving cholesterol efflux and neuronal growth and repair [34,35]. In contrast, the APOE4 variant is associated with an increased risk of AD, playing a significant role in the onset of early Aβ formation and aggregation [35,36,37,38]. APOE4 also plays a significant role in Aβ clearance, interfering with endosomal-lysosomal degradation via downregulation of Rab35 [39,40] and potentially also affecting tau clearance [41].

Although APOE4 was found to be significantly associated with memory function in non-dementia elderly individuals [19], APOE genotyping remains the most common risk factor utilised to test for sporadic AD risk, and in clinical diagnosis [11,42,43]. Over the last several decades, additional genetic variants have been found to be associated with the hallmarks of AD pathophysiology and the cellular action of APOE, which may also be considered as potential key genetic biomarkers for the sporadic form of AD.

### 2.2. Candidate Genes Involved in the Regulation of AD Pathways

One of the most significant genes associated with sporadic AD after APOE4 is Bridging integrator 1 (BIN1) according to GWAS studies (*p* < 3.92 × 10^−58^ [44] and 3.38 × 10^−44^ [45]) and gene ranking prioritisation studies (Table 2). BIN1 is highly involved in lipidic membrane trafficking, endocytosis, and cytoskeleton regulation through its SH3 domain interacting with tau [46,47,48]. BIN1 variants (such as rs744373 and rs59335482) are associated with increased tau (PET-tau, CSF t-tau, CSF p-tau), and subsequent memory deficits [46,49]. A posttranslational modification of BIN1 (phosphorylation in the CLAP domain at T348) also results in increased interaction between BIN1 and tau [46]. Neuronal BIN1 also plays a role in APP processing by reducing RIN3-dependent β-secretase APP processing which leads to decreased Aβ generation [50]. The SNP rs744373, which results in tau accumulation and abnormal Aβ levels as detected by CSF testing [49], was significantly associated with the sporadic form of AD in Caucasian populations, but not in East Asian populations [51]. Interestingly, BIN1 was significantly associated with decreased Age of Onset (AOO) for the sporadic form of AD (*p* = 1.21 × 10^−19^) [52], suggesting its potential utility in diagnostic testing and as a target for development of therapeutics.

**Table 2 ijms-24-13480-t002:** Candidate genes ranking for sporadic AD.

Ranking	Gene	*p*-Value	Source
1	BIN1	2.06 × 10^−30^	[18]
		1.10 × 10^−54^	[53]
		7.30 × 10^−49^	[54]
2	PICALM	4.29 × 10^−14^	[18]
		5.21 × 10^−26^	[53]
		5.10 × 10^−36^	[54]
3	CLU	6.60 × 10^−13^	[18]
		7.71 × 10^−26^	[53]
		1.10 × 10^−28^	[54]
4	CR1	1.51 × 10^−14^	[18]
		1.40 × 10^−23^	[53]
		1.60 × 10^−28^	[54]
5	MS4A	2.87 × 10^−20^	[18]
		9.33 × 10^−20^	[53]
		1.10 × 10^−18^	[54]
6	TREM2	2.34 × 10^−11^	[18]
		1.83 × 10^−23^	[53]
7	PILRA	7.41 × 10^−10^	[18]
		3.28 × 10^−18^	[53]
		1.10 × 10^−18^	[54]
8	SORL1	1.76 × 10^−8^	[18]
		5.59 × 10^−14^	[53]
		4.80 × 10^−17^	[54]
9	HLA	2.88 × 10^−15^	[53]
		1.20 × 10^−14^	[54]
10	CD2AP	2.95 × 10^−12^	[18]
		1.11 × 10^−11^	[53]
		5.80 × 10^−14^	[54]
11	ABCA7	2.36 × 10^−9^	[18]
		2.41 × 10^−13^	[53]
		3.70 × 10^−10^	[54]
12	SLC24A4	3.55 × 10^−7^	[18]
		7.45 × 10^−14^	[53]
		1.10 × 10^−10^	[54]
14	CLNK/EPHA1/ECHDC3/HS3ST1	5.02 × 10^−8^	[18]
		1.08 × 10^−11^	[53]
		3.40 × 10^−7^	[54]
15	ADAM10	2.67 × 10^−11^	[53]
		5.50 × 10^−11^	[54]
16	CASS4	1.56 × 10^−6^	[18]
		1.07 × 10^−10^	[53]

Ranking based on *p*-values reported in various GWAS for AD [18,53,54]. *p*-values refer to the significance of the association between the genetic loci and AD genetic score risk calculations.

After *APOE* and BIN1, GWAS have identified the Phosphatidylinositol-binding clathrin assembly protein (*PICALM*) gene as the most significant genetic susceptibility locus for AD (Table 2) [55]. PICALM initiates clathrin-mediated endocytosis and the autophagy neuronal process [55]. *PICALM* additionally regulates the internalisation of LRP1 (low-density lipoprotein-receptor-related protein-1), β-secretase (BACE1) and γ-secretase expression [55]. The *PICALM.4* isoform is reported to have increased expression in early AD, contributing to the accumulation of endosomal-contained immature protease cathepsin D (CTSD) and reduction of Aβ clearance [56]. In addition, the T/A allele at rs3851179, located in the upstream region of *PICALM,* also provided a protective effect against AD, associated with increased Aβ clearance, lower ratios of CSF p/t-tau [57], and a slower rate of atrophy of the hippocampus [58]; however, the non-protective C allele in rs3851179 has been associated with early AOO of sporadic AD [58]. Interestingly, PICALM is able to mitigate effects caused in the cell from *APOE4*, suggesting a possible regulatory interaction between the two genes [59].

ATP-binding cassette, sub-family A, member 7 (ABCA7), has also been reported to regulate APOE assembly into high-density lipoproteins, as well as controlling glial cell states, leading to a potential protective role in AD [60]. In a Chinese cohort, SNPs in ABCA7 have shown a strong association with AD. The G allele of rs3764650 increased AD susceptibility while the A allele in rs4147929 led to higher morbidity [61]. Loss-of-function variants in protective ABCA7 result in significantly increased AD risk (0.2-fold increased risk in African ethnicity, 1–4 fold increased risk in European origin) [62,63].

Recent studies showed evidence that Triggering receptor expressed on myeloid cells 2 (TREM2) may be a putative receptor for APOE, suggesting a potential role in AD pathogenesis [64]. TREM2 is cleaved by ADAM17 and ADAM10 (significant α-secretases in the brain) to produce soluble TREM2 (sTREM2). Cleavage by γ-secretase produces DAP12, used for signalling [65]. Although some ADAM10 variants have been associated with higher susceptibility to develop the sporadic form of AD [66], a clear genetic correlation of α-secretase haploinsufficiency with clinical symptoms remains to be established; however, variants in TREM2, such as R47H and R62H, have been associated with decreased metabolic function (glycolysis, ATP levels, mTOR activation), leading to a reduction of Aβ toxicity and Aβ plaque formation [65]. Caucasian individuals carrying the R47H variant have an odds ratio for the development of AD similar to that of individuals carrying the significantly more common APOE4 allele [65]. Interestingly, TREM2 is also involved in two hallmarks of AD pathophysiology: hyperphosphorylation of tau and microglial activation [67].

Myeloid cell surface antigen CD33, expressed on the surface of microglial cells, along with CR1 encoding complement receptor 1 (CR1), could play a role in AD by modulating microglial activation [68]. CR1 dysfunction decreases amyloid clearance and leads to improper activation of C3b, leading to synaptic damage [69]. CR1 has 4 alleles, with the second allele (CR1*2) increasing AD risk by 30% [69]. Several CR1 variants (rs4844609 and rs6656401) and (rs2274567) were associated with AD patients in large Caucasian and Latin American cohorts, respectively [69,70]. Not all CR1 variants are causative of AD, with the rs17259045 being found to be associated with decreased Aβ accumulation (detected by increased CSF-Aβ42) in AD patients. Importantly, the rs12567945 variant has been seen to do the same in healthy controls [71]. Another complement protein, clusterin (CLU), has emerged to play a critical role in AD pathophysiology due to its involvement in lipidic transport, neuronal apoptosis, and oxidative stress [69]. Association between the CLU variant, rs11136000, with hallmarks of AD in imaging studies [72], has been reported in AD GWAS [73,74] and case-control association studies [74,75].

GWAS also identified CD2 associated protein (CD2AP), encoding for a neuronal protein essential for signal transduction and cytoskeleton function, as a gene highly associated with the sporadic form of AD (*p* = 1.70 × 10^−17^) [44]. The SNP rs9296559 in CD2AP was found to be strongly associated with CSF p-tau and t-tau levels in early stages of AD [76]. In addition, CD2AP expression was reported to be strongly positively associated with the Braak neurofibrillary stage [77], confirming a role for this gene in AD pathogenesis.

While GWAS have been essential in establishing the foundational genetic architecture of AD, the prioritization of genes identified in these GWAS take into account the regulation of AD susceptibility, candidate causal genes related to the underlying molecular pathways of AD, network analysis, quantitative trait loci, and AD pathophysiology [18,53,54]. Table 2 reports the ranking of key genes discussed above and reinforces the importance of prioritizing genes that modulate AD susceptibility tested in peripheral tissues. Interestingly, previous studies utilizing an algorithm to identify and rank “driver genes” from GWAS results and gene expression studies in peripheral and brain tissues shared 40% of their top 20 ranked genes with genes reported in Table 2 [78], confirming the potential of these ranked genes as promising diagnostics. Although more research is required to reach full validation for use as a diagnostic tool, these top genes presented in Table 2 possess significant roles in the pathways underlying AD pathophysiology, and are key targets for continuing research in AD.

Although GWAS are valuable and contribute to the characterisation of AD genetic markers and identification of genes modulating AD risk, it is essential to consider that 90% of the genome is non-coding [79]. Genetic variants in noncoding regions play an essential role in gene regulation [80], and may exert phenotypic effects via the perturbation of transcriptional gene promoters and enhancers [81]. Epigenetic mechanisms, such as DNA methylation and transcriptome regulation (e.g., the role of miRNAs at the post-transcriptional level), are increasingly recognized to play an important role in the aetiology of AD [82,83]. Current evidence shows that major transcriptome dysregulation occurs in AD pathogenesis leading to changes in the expression of different RNA species, including microRNAs (miRNAs), circular RNAs (circRNAs) and long non-coding RNAs (lncRNAs), which could be used potentially as early biomarkers for AD.

## 3. RNA Regulation

RNA regulation, including of non-coding RNAs, plays an essential role in the modulation of the transcriptome in healthy and pathophysiological contexts. In neurodegenerative diseases such as AD, understanding this context may lead to new avenues of diagnosis and therapy. Three classes of non-coding RNA, miRNAs, circRNAs and lncRNAs, have recently gained greater attention, particularly for their regulatory functions in multifactorial diseases including AD pathogenesis [82,84].

### 3.1. Micro-RNAs

MiRNAs are small non-coding RNAs, 18–25 nucleotides long, able to bind to complementary sequence elements in the messenger RNA (mRNA) of protein-coding genes (“target genes”). This binding, specifically occurring in the 3′ untranslated region, may result in translational repression or degradation of target messenger RNAs (mRNAs) via the recruitment and action of micro-ribonucleoprotein complexes [85]. Alterations in miRNA expression profiles have been reported in AD pathogenesis, impacting key AD-associated pathways and processes, including both amyloid and tau pathologies.

In a similar convention to the genetics of AD discussed previously, a myriad of miRNAs has been implicated in AD pathogenesis (summarised in Table 3, which reports miRNA regulation in AD in peripheral tissues). Recent meta-analyses have identified top miRNAs that are differentially expressed in the brain and other tissues in AD patients compared to healthy controls [86,87]. Among these miRNAs, a role for miR-9-5p in AD pathogenesis has been reported in several studies. MiR-9-5p has been implicated in regulating genes that modulate synaptic plasticity and amyloid plaque accumulation by targeting BACE1, Sirtuin 1 (SIRT1), and Calcium/calmodulin-dependent protein kinase 2, CAMKK2 mRNAs [88]. MiR-9-5p has been identified as significantly contributing to protecting the brain from neuroinflammation and oxidative stress [89], suggesting that it could make a relevant candidate for future AD therapy. Although this miRNA was found downregulated in the brain in several studies, miR-9-5p was also reported to be downregulated in the blood of sporadic AD patients, indicating its role as a possible biomarker for the disease [90].

In addition to the above, numerous miRNAs have been recently underlined as potential biomarkers for the disease, including several studies exploring panels of miRNAs to distinguish between healthy individuals and those with AD and other forms of mild cognitive impairment (MCI). Since Leidinger utilised a Next Generation Sequencing (NGS) approach to identify and validate a panel of twelve blood-based miRNAs (see Table 3), several other studies have presented similar miRNA panels, validating nine micro-RNA in serum samples, including miR-29c-3p and miR-19b-3p as candidate biomarkers for AD [91]. Recently, NGS methods have also allowed researchers to identify miRNA differentially expressed in plasma samples from AD patients and healthy controls [92]. Seven miRNAs (miR-9-5p, miR-29a-3p, miR-106a-5p, miR-106b-5p, miR-107, miR-125a-3p, and miR-125b-5p) were located in the upstream genomic regions of the AD related genes, APP, PSEN1, PSEN2, and BACE1 [93], providing insight into miRNAs as biomarkers, while also linking them to genes implicated in AD.

Guevremont utilised a concurrence approach in plasma with several cohorts to identify miRNAs that were consistently dysregulated throughout disease progression, namely miR-195-5p and miR-324-5p [94]. Both of these microRNAs were confirmed as relevant in AD by other studies (see Table 3). A six-miR panel, including miR-200a-3p, miR-502-3p and miR-142-3p, was reported as differentially expressed in AD, as well as influencing target genes implicated in AD pathology [95]. When looking at miRNA profiling in CSF from case-control AD cohorts, along with genetic variants in AD genes, one study found that three miRNA combinations, with the addition of APOE4 allelic status, were able to effectively (84%) discriminate between AD and controls [96].

A recent meta-analysis of 25 case-control studies identified six miRNA (miR-26b-5p, miR-615-3p, miR-4722-5p, miR23a-3p, and miR-27b-3p) to be differentially expressed in AD in various tissues (including peripheral) [97]. Another study utilised a machine learning approach to identify a 12-miRNA panel able to differentiate between controls and AD participants in serum [98]. Sixteen upregulated and five downregulated miRNAs were reported differentially expressed in serum from a large AD case-control dataset. These were implicated in various AD-related pathways (Mitogen-activated protein kinase, MAPK signalling pathway) via Kegg and GO enrichment analysis [99].

The association of miRNAs and cognitive measures in patients is a key area of interest to reinforce biomarker candidates in clinical settings. One study identified miR-331-3p to be downregulated in AD patient serum, correlating with Mini Mental State Examination (MMSE) score as well as inflammatory marker levels [100]. Similarly, plasmatic miR-107 and miR-103 were reported to be positively correlated with MMSE and negatively correlated with severity of dementia in AD patients [101].

Functional analysis of miRNAs identified as biomarker candidates and their regulatory effects on genes and pathways implicated in AD pathogenesis has also been recurrently explored. One such biomarker, miR-384, exhibits reduced expression in the blood, serum, and CSF of patients with AD, while over-expression of the miRNA is implicated in the suppression of APP and BACE1 expression [102].

**Table 3 ijms-24-13480-t003:** Regulation of RNA dynamic in peripheral tissues in AD pathophysiology.

Downregulation			
RNA Name	RNA Type	Sample	Refs.
miR9–5p, miR-106a-5, miRNA-106b-5p, miRNA-107, miRNA-103a-3p, miRNA-107, miRNA-532–5p, miRNA-26b-5p, let-7f-5p	miRNA	Blood	[88,89,90,93,103]
miR-125a-3p, miR-223p, miR-24–3p, miR-6131, mi-125b-1–3p, miR-331–3p, miR-30b-5p, miR-142–3p, miR-223, miR-29c-3p, miR-19b-3p	miRNA	Serum	[91,95,99,100,104,105]
miR-34a, miR-125a, miR-146a; miR-103, miR-107; miR-135a, miR-193b, miR-384, miR-342–3p, miR-141–3p, miR-342–5p, miR-23b-3p, miR-125b-5p, miR-24–3p, miR-152–3p, miR-107 and miR-103	miRNA	Plasma	[101,102,106,107]
ssssmiR-15a-5p, miR-210, miR-27a-3p, miR-34a, miR-125b, miR-146a, miR-143–3p, miR-142–3p, miR-328–3p, miR-193a-5p, miR-19b-5p, miR-30d-5p, miR-340–5p, miR-140–5p, miR-125b-5p, miR-223–3p, miR-384	miRNA	CSF	[96,102,106,108,109]
			[91]
circ_0089894, circ_0003391	circRNA	Blood	[110,111]
Circ_PCCA, circ_HAUS4, circ_KIF18B	circRNA	CSF	[112]
circRNA_104395, circRNA_402904, circRNA_403472	circRNA	PBMC	[113]
LOC107987206, LOC105378179	lncRNA	Blood	[114]
**Upregulation**			
**RNA Name**	**RNA Type**	**Sample**	**Refs.**
miR-112, miR-161, let-7d-3p, miR-5010–3p, miR-26a-5p, let-7f-5p, miR-1285–5p, miR-151a-3p	miRNA	Blood	[103]
miR-6761–3p, miR-6747–3p, miR-6875–3p, miR-6754–3p, miR-6736–3p, miR-6762–3p, miR-6787–3p, miR-208a-5p, miR-6740–3p, miR-6778–3p, miR-6753–3p, miR-6716–3p, miR-4747–3p, miR-3646, miR-595, miR-4435 miR-28–3p, miR-128, miR-455–3p, miR-483–5p, miR-486–5p, miR-200a-3p, miR-142–3p	miRNA	Serum	[95,99,115,116,117]
miR-29a, miR-29b, miR-195-5p, miR-324-5p	miRNA	Plasma	[94,106]
miR-29a, miR-29b, miR-4449, miR-1274a, miR-4674, miR-106a, miR-24–3p, miR-99b-5p, miR-124–3p, miR-125a-5p, miR-223–3p, miR-140–3p, miR-30a-5p, miR-30e-5p, miR-613, miR-384	miRNA	CSF	[96,102,118,119,120,121]
circ_0077001, circ_0022417, circ_0014356, circ_0014353, circ_0074533, circ_RS-7 or CDR1,	circRNA	Blood	[110]
circLPAR1, circAXL, circGPHN	circRNA	CSF	[112]
circRNA_103366, circRNA_103936, circRNA_101618, circRNA_405619, circRNA_000843	circRNA	PBMC	[113]
lncRNA BACE1	lncRNA	Plasma	[122]
LINC02067, LINC00987, NORAD, ANKRD34C-AS1, THCAT158	lncRNA	Blood	[114]

Although microRNAs are of high interest as potential biomarkers for the sporadic form of AD due to their regulatory role in critical AD genes, more research is warranted to fully consider the use of specific micro-RNA as a diagnostic tool. In addition, emergent non-coding RNA molecules, highly expressed in several body fluids, have shown the potential to significantly regulate the transcriptome.

### 3.2. Emerging Regulators of the Transcriptome: circRNAs and lncRNAs

CircRNAs are a type of noncoding RNA characterised by a covalent closed-loop structure that differentiates them from other noncoding RNAs, such as lncRNAs and miRNAs. Over 180,000 circRNAs were recently found to be present in human transcriptomes, with their expression being associated with both healthy and pathological conditions [123,124]. Although still being explored, the role of circRNA can occur by acting as miRNAs or RNA binding protein sponges to regulate target gene expression, regulating gene splicing, and acting as templates for protein translation in multiple biological processes [125,126]. Due to the tendency for circRNAs to accumulate during healthy brain aging, they may be considered to be suitable markers or treatment candidates for age-related neurodegenerative diseases, such as AD [127].

CircRNAs can affect the expression of many protein-coding genes including *APP* and *BACE1*, which are involved in the regulation of amyloid production. For example, the expression of circRNA ciRS-7, which originates from cerebellar degeneration-related protein 1 antisense transcript (CDR1AS), leads to the activation of Ubiquitin C-Terminal Hydrolase L1 (UCHL1), which then promotes APP and BACE1 degradation, consequently leading to impeded amyloid plaque growth [128]. In addition, the downregulation of ciRS-7 and its “sponging” effects on miRNA-7 induce an increase of miR-7 levels [129], leading to the downregulation of AD-associated targets, such as ubiquitin protein ligase, UBE2A, an essential enzyme in the process of amyloid clearance in AD brain [130]. The circ_HDAC9/miR-138/Sirt1 pathway was reported to play a role in mediating synaptic function and APP processing in an animal model of AD [131]. Such miRNA-mRNA regulatory systems may represent another crucial aspect of epigenetic control over gene expression in health and disease. Inhibition of “miRNA sponging systems” by circRNAs and an increase of specific inducible miRNAs might be a reason for downregulation of important genes related to sporadic AD in the brain. CircRNAs, such as circ_APP and circ_Sirt1, can also influence epigenetic modifications, including DNA methylation and histone modifications, contributing to AD-associated epigenetic dysregulation.

Although some studies have identified a six-circRNA panel that differentiated between AD patients and controls (see Table 3), with the potential ability to discriminate between AD patients and other types of dementias [110], a limited number of studies have explored circRNAs in peripheral tissues in the context of AD. The expression of both circ_0003391 and circ_HAUS4 were positively correlated with several cognitive tests including MMSE, while the expression of circ_GPHN and circ_AXL were negatively correlated with MMSE [111,112]. Interestingly, both circ_GPHN and circ_AXL were also found to be negatively correlated with tau and Aβ42 levels, confirming a potential biomarker role for AD. In addition, circ_PCCA expression was reported to be decreased in the CSF of AD patients when compared to healthy controls [112]. Wang et.al. suggested that circ_PCCA may bind to miR-138–5p leading to the inactivation of glycogen synthase kinase-3β and facilitating tau phosphorylation. Consequently, low circ_PCCA expression may be a good biomarker for illustrating more severe stages of AD [132]. Although the potential role of these circRNA as AD biomarkers is promising, further replication studies in larger cohorts are necessary.

Another subclass of non-coding RNAs, lncRNAs, have also emerged recently as significant regulators of the transcriptome. Over 50,000 human lncRNAs have currently been identified [133]. These molecules have at least 200 nucleotides and can interact with DNA, RNA, and RNA-binding proteins at the transcriptional, post-transcriptional, and post-translational level [134]. Most lncRNAs are found in the nucleus and play a key role in the regulation of gene expression and translation via interactions with DNA, mRNAs, proteins, and miRNAs [135]. Increasing evidence suggests that aberrant expression of lncRNAs correlates with AD progression. LncRNAs have been associated with different aspects of AD pathology, such as regulation of Aβ peptide, tau, inflammation, autophagy and neuronal cell death [136].

Highly expressed in the brain and the periphery, these lncRNAs are often dysregulated in patients with neurological disorders, including AD [137]. Dysregulation of lncRNAs have been reported in plasma from AD sufferers, such as lnc_BACE1 [122] and in the blood including LINC02067, ANKRD34C-AS1, and THCAT158 (Table 3) [114]. These three lncRNAs have been previously reported to be associated with AD in a LncRNA Disease v2.0 Database, but their specific role in AD pathophysiology remains unknown [138].

It is important to note that the dysregulation of non-coding RNA in AD is complex and evidence supporting their roles in AD pathophysiology is relatively recent, and further research is required to fully uncover the mechanisms and functional implications for AD pathophysiology.

## 4. Discussion

Pre-clinical use of genetic testing can provide an early diagnostic tool for AD, investigating not only the key hallmarks of AD (as often tested via invasive CSF lumbar puncture procedures and expensive, multi-day PET scan procedures to visualize early abnormal protein accumulations), but also the subtle pathological drivers of AD, including microglial dysfunction, deficient repair and clearance or damaged cells, dysfunctional complement activation and impaired cerebral vasculature. Taken as a whole, preliminary genetic testing using low-cost and non-invasive peripheral sampling, including blood and saliva, can provide a more comprehensive picture of these early stages of AD. This is critical, as novel frontline treatments for AD currently function to decrease and subdue the formation of key AD pathologies (such as amyloid-beta plaques and tau neurofibrillary tangles) but cannot repair any damage already done by this devastating pathology.

Many of the genes identified via genetic studies and discussed here are involved in more than one pathway underlying AD pathophysiology (such as APOE, PICALM, CLU, CD2AP, TREM2, BIN1, and reported as prioritised genes in Table 2), supporting the fact that disease and therapy should be considered convergently instead of in isolation. A multi-genic approach, along with the assessment of additional risk factors involved with these genes, must be considered during the development of a potential diagnostic tool. This report also highlights the significance of RNA molecules regulating the transcriptome, particularly related to expression of genes involved in AD pathophysiology. Due to their effects on target genes in AD pathways, miRNAs have been highly studied in the last decade for their diagnostic and therapeutic potential [139]. Although miRNAs are very targeted molecules, they need to pass through the brain–blood barrier to effect gene expression in the brain and only a low number can get through even if administered in high doses systemically. Consequently, their major potential may be for diagnostic purposes. Considering that all RNA molecules are present in both brain and peripheral tissues (CSF, blood), they have the potential to assist AD prognosis.

CircRNAs should also be considered when establishing a potential “diagnostic profiling” test of the most effective genetic molecules able to reflect AD diagnosis and disease progression due to their “sponging” effects on mi-RNAs, effecting their regulation, and considering the high stability of these molecules in body fluids (covalently closed ends endow high resistance to enzymes).

Due also to the ability of lncRNAs to modulate cellular autophagy and their role on BACE1, targeting autophagy-associated lncRNAs in neuronal cells may be a new avenue for therapeutic support for AD [122]. In addition, the finding that a growing number of lncRNA have been found to be associated with the prognosis of patients suffering from cancer such as hepatocellular carcinoma [140], suggests their relevance in the context of AD diagnosis should be further studied. Furthermore, miRNAs are also considered as biomarkers for some types of cancer due to their altered miRNA expression profiling reflecting disease development [141] and studies have confirmed a potential role of exosomal circRNAs in tumour cell proliferation, metastasis, drug resistance, and progression [142]. Similarly, recent studies also investigated the role of miRNA and circRNAs as potential peripheral biomarkers for neurodegenerative diseases (including Parkinson’s Disease), but replication studies in larger cohorts remain to be performed [143,144].

Some limitations must be considered before considering these genetic markers for clinical use as AD peripheral biomarkers. The disparity of results due to different methods of detection must be streamlined, utilizing better universal technology. In addition, the results must be validated using large cohorts of AD case-control participants to further verify the clinical value of these genetic markers as AD diagnostic biomarkers.

While this review provides a summary of ranked genes involved in the genetic susceptibility of AD development, the characterisation of a single, definitive genetic test to predict AD remains incomplete. Significant additional research is needed to both characterize AD genetic variants of interest and evaluate their role in an individual’s genetic vulnerability towards the development of sporadic AD. Taking into account that the large majority of our genome is non-coding [79], exploring the regulatory role of this non-coding part of the genome beyond coding regions (such as AD ranked genes seen in Table 2) is critical. Consequently, a multi-genetic approach which includes regulatory non-coding RNA profiles (i.e., including miRNAs, lncRNAs and circRNAs) may offer a promising avenue for clinical diagnosis for AD. The use of miRNAs as prognostic biomarkers has already been considered as clinically relevant in oncology. For example, significant associations have been reported for predisposition to develop colorectal cancer, lung cancer, breast cancer and pancreatic cancer with increased blood level of miR-21 [145,146,147,148,149]. One of the most established lncRNA, PCA3, has been approved for use in the diagnosis of prostate cancer and is currently being explored as a therapeutic target [150]. Highly stable circ-RNA such as hsa_circ_002059, may be a potential novel and stable biomarker for the diagnosis of gastric carcinoma [151]. With no valid blood biomarkers for lung cancer, circRNA_100876, hsa_circ_0014130 and circFARSA were characterised as promising biomarkers, particularly for non-small cell lung cancer (NSCLC) prognostic [152,153,154]. Altogether, identification of non-coding RNA markers associated with early stage of AD appear to be promising for use as a prognostic tool. However, detection methodologies as well as biological knowledge gaps must be addressed in sporadic AD research before moving this emerging field of research forward and bring non-coding RNA to the forefront of clinical practice.

Most of the studies utilised for GWAS and non-coding RNA studies are performed in body fluids from AD patients with severely expressed symptoms (late stage of disease). For example, BIN1 genetic and epigenetic variation (SNP and CpG methylation) were reported to be associated with AD susceptibility in human postmortem temporal lobes [155]. Similarly, methylation at Illumina probe cg02308560 in the *ABCA7* locus was also found to be associated with AD pathology in the same study, confirming the potential of the prioritised genes established at peripheral levels [155]. Interestingly, BIN1 genetic and epigenetic variations were also reported associated early stage of AD in a human brain case-control small cohort [155]. Taking into account that prioritised AD genes are under epigenetic modulation (including RNA non-coding regulation) and RNA dynamics changing with age [156], further exploration of potential genetic markers at the early stage of the disease is critical prior to its potential utilisation as AD prognostic tool. Recent studies have shown that there is a concordance of genome-wide epigenetic changes across three commonly used peripheral tissues (saliva, blood, CSF) with epigenetic patterns reported in live human brain tissue, and a higher correlation between the epigenetic profiles in saliva and brain tissues when compared to blood [157,158], reinforcing the potential for saliva genomic markers to be utilised as an early diagnostic tool for AD.

Consequently, future studies may include the exploration of variants in the top ranked genes (Section 2) and expression of non-coding RNAs (Section 3) in saliva samples from early-stage AD individuals with limited symptoms. Due to the effects of tissue specificity of genetic mutations and RNA non-coding regulation, it is essential to establish a valid tool to assess the presence of mutations in coding and non-coding regions (affecting gene regulation) of the genome in specific type of human body fluid, for example, in saliva, as previously indicated [158]. Advancements in pathology testing have led to the exploration of AD biomarkers in diverse body matrices, such as blood, saliva, urine, tears and olfactory fluids [159]. Taking into account the coding and non-RNA variations seen in different tissues, and the lack of standardized assays for exploring non-coding RNA markers, the use of such biomarkers in clinical practice remains challenging, and a key area for future research.

In conclusion, peripheral miRNAs, circRNAs, and lncRNAs related to AD genes and pathophysiology have significant potential to be further considered as a convergent diagnostic panel biomarkers for AD. Considering these together, these, along with other genetic markers, may be more powerful and provide higher efficacy in the prediction of AD, at the different stages of the disease; however, research on RNA regulatory processes in AD (particularly for circRNA and lncRNA) is still in its infancy and an important area to pursue. Additional large-scale human genetic studies focussing on analysis of body fluids at the early stage of the disease are also warranted, along with the use of an appropriate methodology to validate the findings.

## Figures and Tables

**Figure 1 ijms-24-13480-f001:**
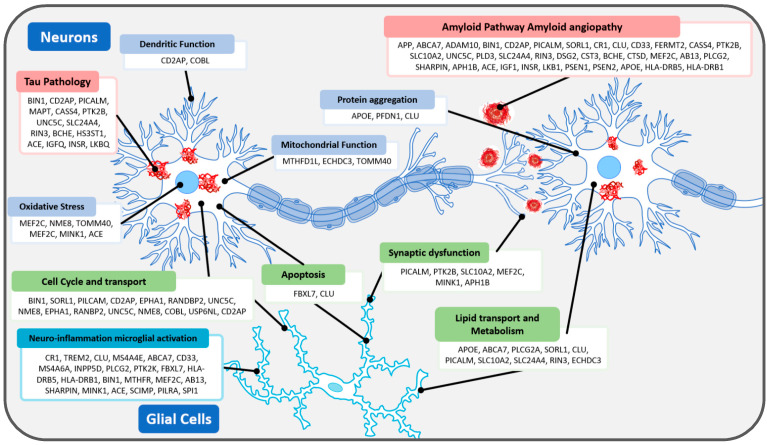
Genes involved in neuronal dysfunction in AD pathophysiology.

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
