# Peer review of "Can Genetic Markers Predict the Sporadic Form of Alzheimer’s Disease? An Updated Review on Genetic Peripheral Markers"

_ijms, 2023, doi:10.3390/ijms241713480_

Round 1
Reviewer 1 Report
Upon reviewing the manuscript entitled ”Can genetic markers predict the sporadic form of Alzheimer’s disease? An updated review on genetic peripheral markers”, it is evident that it lacks a firm conclusion and does not provide a clear path for future directions. While the paper delivers several interesting and valid points, these ideas are left hanging without definitive endings or explanations. Future directions are vital as they provide subsequent researchers with leads to follow, opening new venues for discovery. Without clear conclusions and guidelines for future research, the paper falls short of its scientific aims and fails to realize its potential fully.
Another striking shortfall of the work is its lack of prioritization or ranking of markers. Given the wide range of markers typically involved in clinical research, providing a ranking or prioritization based on the cost-effectiveness or efficacy of detection is critical. Without this, readers are confused about the best markers to use in their research or clinical practice, limiting the paper's utility. Including such a ranking or prioritization system would improve the paper's usability and add depth to the investigation. The findings must be relatable to clinical practice for the article to be helpful in a practical setting. Without this, the results remain purely academic and do not translate into beneficial applications in real-life clinical scenarios.
Overall, while the paper presents significant research and findings, the noted shortcomings significantly diminish its potential impact on the field. For the future, the authors should strive to provide clear conclusions, define future research directions, and ensure a strong focus on clinical relevance. Additionally, including a prioritization system or ranking markers would significantly enhance the paper's usability and depth. By addressing these issues, the authors can substantially improve the quality and impact of their research.
Author Response
Please see attachment "response to reviewers" and the new tracked version of revised manuscript.
Thank you

Reviewer 2 Report
This is a well-written narrative review; the authors present an updated picture of the candidate genes that could be used for early diagnosis of late-onset Alzheimer’s disease (AD). Almost 60-80% of sporadic AD can be linked to genetic factors. Therefore, there is a large and growing body of research on the genetic causes of AD to effectively harness therapies aimed at reducing the pathogenic protein load, early diagnosis of AD should be targeted. The authors aim at discussing genes that could be used for AD diagnosis through samples obtained from peripheral tissues and describe the RNA dynamics that could support early detection.
While the review raises an important question regarding harnessing early diagnosis methods to improve the quality of life of patients with AD, the following points need to be clarified:
General Comments
1. Since this is a narrative review, the authors should consider discussing each topic under its respective title. This would provide more flow to the manuscript. The authors could consider including a short conclusion at the end of the manuscript for the key messages.
2. Genetic testing is not prescribed for late-onset Alzheimer’s disease. The authors should also discuss how genetic testing could be prescribed to people who may not be displaying any AD symptoms for early diagnosis. This would provide additional rationale for further studies on diagnostic biomarkers for late-onset sporadic AD.
3. Regarding the potential genes which could be candidate markers for early detection of AD, the authors should consider tabulating them along with their normal physiological function, expression in peripheral tissues vs. central nervous system, and changes concerning AD. Based on the available evidence, the authors could also suggest which genes could be the most prominent diagnostic candidates for AD.
Specific Comments
1. Line 195-197: The reference cited here does not talk about the GWAS studies in AD, please clarify.
2. Line 373-374: Please include a reference for the statement
3. Line 378-384: The author should consider citing more research done with lnRNAs, circRNAs, and miRNAs should be studied concerning AD or neurodegenerative disorders and how can they be levered as diagnostic markers as in some types of cancer.
Author Response
Please see attachment and the modified version of the manuscript with tracked changes
Thank you

Round 2
Reviewer 1 Report
Based on responses to previous comments, I conclude for acceptance in the present form.